# Pushing the envelope: Micro-transmitter effects on small juvenile Chinook salmon (*Oncorhynchus tshawytscha*)

**A. Michelle Wargo Rub**[1]*, **Benjamin P. Sandford**[2], **JoAnne M. Butzerin**[3], **April S. Cameron**[4]

**1** Fish Ecology Division, Northwest Fisheries Science Center, National Oceanic and Atmospheric Administration, Hammond, Oregon, United States of America, **2** Fish Ecology Division, Northwest Fisheries Science Center, National Oceanic and Atmospheric Administration, Pasco, Washington, United States of America, **3** Fish Ecology Division, Northwest Fisheries Science Center, National Oceanic and Atmospheric Administration, Seattle, Washington, United States of America, **4** Ocean Associates, Inc., Hammond, Oregon, United States of America

* michelle.rub@noaa.gov

**Data Availability Statement:** Data that is not contained within the manuscript or supporting information files and can instead be found in the appendices of our contract reports using the

## Abstract

Significant effort has been invested in downsizing telemetry transmitters so they can be used to monitor survival and behavior in a variety of fish species and life stages. Commercially available "micro" transmitters in particular have presented researchers with the opportunity to tag very small fish (< 250 mm fork length). We conducted a release/recapture study in tandem with a laboratory study of tag effects on juvenile yearling spring and subyearling fall Chinook salmon (*Oncorhynchus tshawytscha*). Fish surgically implanted with both a micro-acoustic transmitter and passive integrated transponder (PIT) tags were compared with fish injected with only a PIT tag. Detections from both tag types showed that during the downstream migration, fish surgically implanted with both a micro-acoustic transmitter and PIT tag did not survive at the same rate or behave in the same manner as those injected with only a PIT tag. Differences in survival were more pronounced in subyearlings than in yearlings. This was likely due to warmer temperatures experienced by migrating subyearlings, their higher metabolic rate, and their smaller size and consequently higher tag-burden. To identify the mechanisms driving these differences, we necropsied migrating study fish recaptured at locations 225-460 km downstream from the release site. Results revealed that compared with PIT-tagged fish, micro-acoustic-tagged fish had heightened inflammatory responses within the body cavity, delayed healing of surgical incision sites, and poor body-condition. For study fish tagged along with those released to the river but held in the laboratory for observation, outcomes revealed that tag effects were similar in direction, but not as pronounced under artificial conditions.

## Introduction

Electronic transmitters have been used in fisheries research since the mid-1950s to track survival and movement of many fish species and sizes [1]. Transmitter technology provides scientists the opportunity to remotely monitor fish behavior in the natural environment over space

following links. For 2007, there is an A and B appendix for the yearling and subyearling data, respectively. Links and citations are as follows: https://www.nwfsc.noaa.gov/assets/11/7568_03282014_095347_Wargo-Rub.et.al.2009.Online-Appendix-YCS.pdf and https://www.nwfsc.noaa.gov/assets/11/7569_03282014_101305_Wargo-Rub.et.al.2009.Online-Appendix-SYCS.pdf. Yearling citation: Wargo Rub, A. M., R. S. Brown, B. P. Sandford, K. A. Deters, L. G. Gilbreath, M. S. Myers, M. E. Peterson, R. A. Harnish, E. W. Oldenburg, J. A. Carter, I. W. Welch, G. A. McMichael, J. W. Boyd, E. E. Hockersmith, G. M. Matthews. 2009. Appendix A to Comparative performance of acoustic-tagged and passive integrated transponder-tagged juvenile salmonids in the Columbia and Snake Rivers, 2007. Detection data from acoustic and passive integrated transponder tags for yearling Chinook salmon. Access data via metadata icon. Subyearling citation: Wargo Rub, A. M., R. S. Brown, B. P. Sandford, K. A. Deters, L. G. Gilbreath, M. S. Myers, M. E. Peterson, R. A. Harnish, E. W. Oldenburg, J. A. Carter, G. A. McMichael, J. W. Boyd, E. E. Hockersmith, G. M. Matthews. 2009. Appendix B to Comparative performance of acoustic-tagged and passive integrated transponder-tagged juvenile salmonids in the Columbia and Snake Rivers, 2007. Detection data from acoustic and passive integrated transponder tags for subyearling Chinook salmon. There is only one appendix for the 2008 report, since we only released yearlings. Link: https://www.nwfsc.noaa.gov/assets/11/7570_03282014_104342_Online-Appendix.xlsx. Citation: Wargo Rub, A. M., B. P. Sandford, L. G. Gilbreath, M. S. Myers, M. E. Peterson, L. L. Charlton, S. G. Smith, G. M. Matthews. 2011. Appendix to Comparative performance of acoustic tagged and passive integrated transponder tagged juvenile Chinook salmon in the Columbia and Snake Rivers, 2008. Detection data from acoustic and passive integrated transponder tags. Access data via metadata icon.

**Funding:** NOAA Fisheries, NWFSC, Fish Ecology Division (AMWR, BPS, and JMB) received funding for this study from Portland District, U.S. Army Corps of Engineers, under contract W66QKZ60441152. This agency provided support in the form of salaries for AMR & BPS but did not have any role in the study design, data collection and analysis, decision to publish, or preparation of the manuscript. Funding in the form of salary for ASC was provided by NOAA Fisheries through contract with Ocean Associates, Inc. Ocean Associates, Inc. had no role in the study design, data collection and analysis, decision to publish, or

and time. As such, results of telemetry studies are often used to inform management decisions [2]. However, as informative as these studies may be, interpretation of their results can be complicated. When interpreting results of a given telemetry study, one generally assumes that tagged fish are representative of their untagged cohorts (i.e. behavior and survival will be similar between cohorts). However, there is no completely benign method for attaching transmitters to fish, and many variables can influence the success of an individual tagging experiment, including both biological and physical factors. The most commonly used attachment method in contemporary telemetry studies is surgical implantation within the peritoneal cavity [3]. This method in particular can lead to a number of biological effects that may compromise the validity of inferences and conclusions based on studies of this nature. Observed effects of surgical implantation have included impaired feeding, growth, and swimming ability as well as premature mortality [2,4–6]. Potential biological effects are of special concern for very small fish, as there has been a trend to "push the envelope" with respect to the minimum acceptable fish size for tagging. Biological effects are also an important consideration in tagging studies of fish behavior and survival over longer periods [7–12].

Because of inherent difficulties in recapturing tagged fish after release, much of the literature on evaluation of tag effects and relationships of fish size to transmitter size is based on results from laboratory studies [9–10,12–15]. Laboratory evaluations provide important evidence with which to evaluate effects such as tag loss or failure; however, studies conducted solely within an artificial environment may lead to unrealistic expectations of fish performance and survival. Tag effects can be evaluated more rigorously by observing the performance of fish in situ. Yet to date, few comprehensive field studies have been designed and conducted for this purpose [16–18]. No studies of which we are aware have looked at tagging effects from both a field and laboratory perspective in tandem.

During 2007 and 2008, we conducted a multi-faceted study that encompassed inriver release, post-release detection and recapture, and laboratory holding to examine the effects of surgically implanted acoustic transmitters (AT) and the mechanisms behind these effects. We compared tag burdens ranging from 1.0 to 10.9% in actively migrating fish tagged and released under a wide range of environmental conditions, including variations in temperature and river flow.

## Materials and methods

This research was permitted by the NOAA Fisheries, West Coast Regional Office under the 2004 FCRPS Biological Opinion (permit #20-07-NWFSC-46 and #13-08-NWFSC-46).

### Field studies

Field evaluation of tagging effects was based on direct comparisons of migrating juvenile fish surgically implanted with both an acoustic transmitter and passive integrated transponder (PIT) vs. cohorts injected with PIT tags only. In both study years, acoustic-tagged fish were implanted with transmitters manufactured by Sonic Concepts (Bothell, WA) for the *Juvenile Salmonid Acoustic Telemetry System* (JSATS). In 2007, these micro-acoustic tags measured $16.10 \times 4.10 \times 5.90$ mm and weighed 0.6 g in air; in 2008, they were smaller, measuring $12.0 \times 3.50 \times 5.30$ mm and weighing 0.42 g in air (S1 Table). In both years, we used injectable PIT tags from Biomark, Inc. (Boise, ID, TX-1411SST) with an average length and diameter of 12.48 and 2.07 mm, respectively, and an average weight of 0.1 g in air.

**Study sites.** Field studies were conducted at multiple locations spanning several hundred km within the Columbia River Basin in the U.S., Pacific Northwest (Fig 1). Fish were collected, tagged, and released at Lower Granite Dam on the Snake River (rkm 695; 46.6604˚N,

preparation of the manuscript. The specific role of each author is articulated in the 'author contributions' section.

**Competing interests:** NOAA Fisheries, NWFSC, Fish Ecology Division (AMWR, BPS, and JMB) received funding for this study from Portland District, U.S. Army Corps of Engineers, under contract W66QKZ60441152. This agency provided support in the form of salaries for AMR & BPS but did not have any role in the study design, data collection and analysis, decision to publish, or preparation of the manuscript. Funding in the form of salary for ASC was provided by NOAA Fisheries through contract with Ocean Associates, Inc. Ocean Associates, Inc. Neither the affiliation with the U.S. Army Corps of Engineers nor with Ocean Associates, Inc. altered the authors' adherence to PLOS ONE policies on sharing data and material.

117.4280˚W) and monitored as they migrated downstream to Bonneville Dam (rkm 235; 45.6443˚N, 121.9406˚W). This highly regulated basin afforded an ideal study area for our work [19,20] because six of the seven dams encountered by our study fish were equipped with detection systems that recorded information from PIT-tagged study fish that passed within reading range. These detections were used to estimate both survival and travel time through the system for our study fish. In addition, we used the *separation by code* collection systems (SbyC) at three dams, which allowed us to recover individual study fish based on their unique PIT tag code.

**Fish collection and tagging.** All studies were conducted with juvenile Chinook salmon listed under the U.S. Endangered Species Act [21,22]. Appropriate permits were obtained for fish collection and tagging and all work was conducted in conformance to applicable regulations. We targeted two Chinook salmon life history types: yearling spring and subyearling fall; however, subyearlings were excluded from field studies in 2008. Yearling Chinook salmon return as adults in spring and hold in freshwater for months before spawning in fall. Most juveniles spend a year rearing in freshwater before migrating to sea in spring. During the spring migration season, yearling fish that reach Lower Granite Dam are generally 80-200 mm

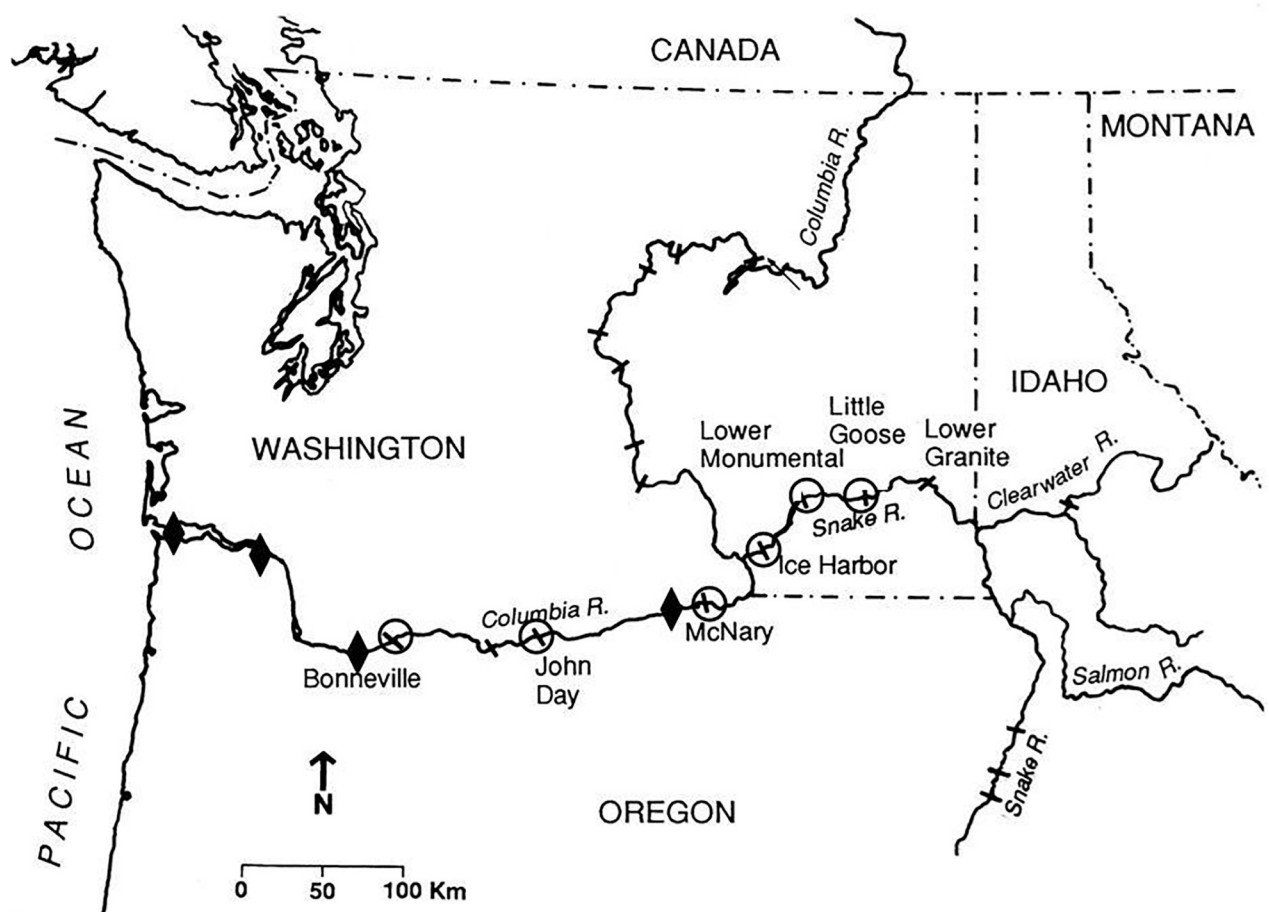

**Fig 1. Columbia River hydropower system, pacific Northwest United States.** Fish were collected, tagged, and released at Lower Granite Dam (46.6604˚N, 117.4280˚W). Diamonds indicate downstream detection sites for acoustic transmitters and circles for PIT tags. *Separation by Code* systems were used to recapture Chinook salmon smolts at McNary Dam (45.9362˚N, 119.2972˚W), John Day (45.7148˚N, 120.6937˚W), and Bonneville Dam (45.6443˚N, 121.9406˚W).

fork length (FL) [23]. Subyearling Chinook salmon return as adults and spawn in fall, with most juveniles migrating to sea during the first year of life, in late spring or summer. Subyearlings generally range 60-140 mm FL by the time they arrive at Lower Granite Dam. Based on our tagging location, study fish were comprised entirely of salmon from the Snake River.

All fish were collected at Lower Granite Dam throughout the juvenile migration period of each life history type. Yearling Chinook included for the study measured at least 95 mm FL and were exclusively of hatchery origin (identified by adipose fin clip). Subyearling Chinook also measured at least 95 mm FL. However, they were comprised of a mixture of hatchery fish and those presumed to be of natural origin (identified by intact adipose fin).

Immediately after collection and sorting, fish selected for PIT-tagging only were sedated with tricaine methanesulfonate (MS-222) [24,25], measured, and injected with PIT tags using a method similar to that of Prentice et al. [26,27]. Fish selected for acoustic tagging were held overnight prior to tagging. The following day, these fish were anesthetized in a bath containing MS-222 in concentrations ranging 50-80 mg/L, until they reached stage-IV anesthesia (loss of equilibrium) [28]. After pre-surgical examination (including weighing and measuring), fish were placed on a surgery table ventral side up, and their gills were irrigated with either MS-222 (50 mg/L), pure river water, or a combination of both. Both a PIT and acoustic tag were inserted into the peritoneal cavity through a ventral midline incision, and each incision was closed with two 5-0 absorbable monofilament sutures placed in a simple interrupted pattern.

Upon recovery from the surgical anesthetic, acoustic-tagged fish were commingled with PIT-tagged fish that had been collected and tagged that day. As such, survival for each acoustic-tag group was compared to that of the PIT group collected approximately 24 h later. Following a post-tagging recovery period of 12-24 h, both treatment groups were released simultaneously into the tailrace of Lower Granite Dam.

**Recapture of study fish.** In order to recapture treatment fish for examination along the migration route, we programmed *separation-by-code* systems to divert the first 10 fish detected from each replicate to collection tanks at up to two sites [19,20]. Yearling fish were recaptured at McNary and Bonneville Dam in spring 2007. We intended to continue this design in 2008, but midway through the study period, high debris loads at Bonneville Dam caused clogging of the fish guidance screens and a subsequent shutdown of the juvenile fish collection system. Thereafter, the balance of our sample fish were recaptured from John Day Dam (112 rkm upstream) and combined with recaptures from Bonneville for analysis.

In summer 2007, subyearling Chinook salmon were recaptured only at Bonneville Dam because the recapture system at McNary Dam was not in service. Our goal was to recapture a maximum of 100 yearlings and 130 subyearlings from each tag treatment group at each downstream collection site (10 from each release group). Recaptured fish were used for gross necropsy and histological examination and to evaluate levels of infection with *Renibacterium salmoninarum* (*Rs*), the causative agent of bacterial kidney disease. To provide a baseline with which to compare the results of these examinations, we also collected a reference group corresponding to each paired release at Lower Granite Dam. Reference fish were anesthetized but not tagged and were sacrificed and necropsied on-site immediately after collection in a manner similar to that of fish recaptured from tag treatment groups.

All recaptured treatment and reference fish were humanely sacrificed within 24 h of collection using an overdose of MS-222 [29,30]. Each fish was measured, weighed, and evaluated for external abnormalities and gross visible injury such as lesions, descaling, or hemorrhaging. Individual necropsies were performed at collection sites in the manner of Noga [31]. Necropsied fish were examined for gross tissue response to tagging, such as tag encapsulation. The following metrics were evaluated using a Goede index scoring system [32]: smolt index, eyes, fins, gills, pseudobranchs, caecal fat, mesenteric fat, spleen, food in stomach, and appearance

of the hind gut, liver, gall bladder, and kidney (Table A in S3 Appendix). Goede index scores were compared between treatments by collection site using a Kruskal-Wallis non-parametric test [33].

Next, tissue samples were taken from the gill, heart, liver, head kidney, trunk kidney, spleen, upper intestine, lower intestine, skin in area of the incision/suture, and pyloric ceca for histological examinations (S4 Appendix). Tissue samples were evaluated based on the histological metrics in Table A in S4 Appendix. After all tissue samples were evaluated, scores were coded and entered into a spreadsheet, and data from each collection site were compared by treatment group using either a chi-square contingency table, a Fisher's exact test (presence/absence data), or a Kruskal-Wallis non-parametric test (ordinal data) [33].

A second kidney sample was collected from each reference and treatment fish at the time of necropsy for enzyme-linked immunosorbent assay (ELISA) as described by Pascho and Mulcahy [34] and modified by Pascho et al. [35]. These samples were taken to determine the presence and levels of *Renibacterium salmoninarum (Rs)*, the causative agent of bacterial kidney disease (S6 Appendix). A Kruskal-Wallis test was used to compare differences in *Rs* antigen levels between treatments and across replicates.

**Estimates of downstream survival and travel time.** For all tag treatment groups, PIT tag detection data was retrieved from *The Columbia Basin PIT Tag Information System* (PTAGIS) and checked for errors [36]. Pre-release mortalities and fish that had lost tags before release were excluded from analyses, as were study fish that had been incidentally removed from the river due to barge transportation or other sampling. For both PIT- and acoustic-tag groups, survival estimates were based upon PIT-tag detections at six hydropower dams downstream from the release site (Fig 1). Survival was estimated for both tag treatment groups using the Cormack-Jolly-Seber model [37–39] implemented using *Survival with Proportional Hazards* software [40]. This method assumes equal probabilities of detection at each downstream monitoring site and equal probabilities of survival for fish belonging to the same treatment group, regardless of prior detection history.

Tag treatment groups were paired by release date at Lower Granite Dam, and comparisons of relative survival were made using paired *t*-tests on the mean and standard error of the ratios of estimated survival between individual pairs of acoustic- vs. PIT-tag replicates (AT/PIT). For each paired group, we evaluated the null hypothesis that survival was equal between tag treatments, that is, the ratio between tag treatment groups (AT/PIT) was equal to one. For these tests, we used log-transformed survival estimates of each ratio. The mean and standard error were then back-transformed to provide estimates on the original scale. Sample sizes were chosen to ensure that from release to approximately 348 km downstream, a minimum survival difference of 5% between groups could be differentiated with 80% power ($\alpha = 0.10$ in 2007 and $\alpha = 0.05$ in 2008).

We estimated the median travel time from release at Lower Granite Dam to each downstream dam for every release and treatment group combination where a minimum of 10 fish were detected. At each downstream detection site, we calculated means and standard errors (SEs) of the medians for all releases by tag type to estimate average median travel time and its variability. Next, the difference in median travel time between tag treatments was calculated for each paired release group and detection location. The average and SE of these differences by location were then used to construct *t*-tests of the null hypothesis that there was no difference in median travel time between tag treatment groups. Detections that occurred 55 d after the tag-activation date (the minimum life of the acoustic transmitters) were excluded from these analyses.

**Smolt-to-adult survival.** Survival to adulthood was estimated for study fish based on PIT tag detections of adult fish at Bonneville in the years following release. Smolt-to-adult return

rates (SARs) were tabulated by release group and treatment. Mean and pooled SAR ratios (AT/PIT) were calculated based on release at Lower Granite Dam as juveniles and subsequent adult detection at Bonneville Dam. Tests of significance were conducted in a manner similar to that used to calculate survival ratios during juvenile migration.

## Laboratory studies

Subsamples from each paired release group were transported to the Bonneville Dam Juvenile Fish Facility, where they were held for extended observation in laboratory tanks. Both acoustic- and PIT-tag treatment fish were obtained from each of the 10 yearling release groups in both 2007 and 2008 and from 9 of the 27 subyearling release groups in 2007. For each laboratory comparison, we collected an additional reference group, which was handled and anesthetized but not tagged.

In 2008, although we did not release subyearlings to the river, we collected and tagged them for laboratory holding on 10 dates over the course of the subyearling migration period. In addition to the acoustic- and PIT-tag replicates and reference groups, we collected fish for a surgically implanted PIT-tag treatment group. Also in 2008, the laboratory holding period was increased from 90 to 120 days. Our goal during both years was to obtain 40 fish for laboratory holding from each treatment and replicate (Table A in S5 Appendix).

All fish assigned to a laboratory treatment were held at Lower Granite Dam overnight after tagging and then transported by truck the next morning to the Bonneville facility. Upon arrival at the facility, fish were transferred (water-to-water) to 1,893-L circular tanks that were maintained with flow-through river water at ambient temperature for 14 d (Table B in S5 Appendix). For all groups, study tanks were converted to a closed artificial seawater system on holding day 15 to mimic ocean entry conditions (35 ppt salinity at 11.1-13°C). Fish were maintained in the artificial seawater system through the remainder of the holding period. Timing of transfer to seawater was based primarily on yearling travel time to ocean entry [41]. For subyearlings, travel time during the summer migration varies considerably [42]; nevertheless, we transferred these groups to seawater at holding day 15 for comparison purposes.

During the entire holding period, fish were fed ad libitum a diet consisting of a mixture of appropriately sized *BioDiet Grower*, a semi-moist pelleted commercial fish food (Bio-Oregon, Longview, WA). Waste food and fish excrement were removed from holding tanks on a continuous basis by self-cleaning flow within the tanks. Fish surviving to the end of the holding period were humanely euthanized with an overdose of MS-222 [29,30], weighed and measured. Kidney tissue was collected from all mortalities and survivors and tested for *Rs* antigen levels as described for fish recaptured at dams (S6 Appendix). Coded-wire tags were also collected from the snouts of individual fish when present, and their respective codes were recorded in a database. These data were reported by [5,6].

We compared post-treatment laboratory survival at day 14 and 28 in both years, and at day 90 and 120 in 2007 and 2008, respectively. For these comparisons we used a two-factor ANOVA Fisher's LSD, with replicate release date as a random factor and tag treatment as a fixed factor. Mean growth in mm (yearling and subyearling) and mean weight gain in g (subyearling) were calculated by replicate for fish from both tag treatments that survived the holding period.

For all fish, tag losses were determined at the time of necropsy. Differences in the percentage of PIT tags lost between tag treatments for spring and summer groups were evaluated using chi-square tests. Kruskal-Wallis tests were used to compare differences in *Rs* antigen levels between treatments and across replicates. Levels of post-mortem *Rs* antigen were compared

among treatment groups for both fish that died prematurely and those that survived the entire holding period.

## Results from field studies

### Downstream survival and travel time

**Yearling Chinook salmon.**    In 2007 and 2008, respectively, a total of 3,818 and 4,139 yearling Chinook salmon were surgically implanted with both an acoustic transmitter and a PIT tag and released to the river (Table 1). Paired replicate groups of yearling fish were injected with PIT tags, with total releases of 46,714 in 2007 and 50,814 in 2008. Mean tag burdens were 3.1% (range 1.5-8.4%) in 2007 and 2.3% (range 1.0-7.2%) in 2008 for acoustic-tagged fish (Table 1 and Fig 2) and approximately 0.4% in both years for PIT-tagged fish, ranging approximately 0.2-1.2% in 2007 and 0.2-1.4% in 2008. Yearling PIT-tag treatment fish were not weighed prior to tagging; therefore, approximate tag burdens are reported based on the weight distribution of fish sampled for the acoustic tag group.

For yearling Chinook salmon released during 2007, survival to Lower Monumental Dam, 106 km from release, was significantly higher for acoustic-tagged fish, with an AT/PIT ratio of 1.05 ($p$ = 0.08; Table 2 and Table A in S1 Appendix). However, by the time fish reached McNary Dam, 225 km from release, survival was higher for PIT-tagged fish (AT/PIT ratio 0.92, $p$ = 0.054; Table 2 and Table B in S1 Appendix). As fish moved further downstream, the survival advantage to PIT groups increased, with AT/PIT ratios of 0.72 at John Day Dam ($p$ = 0.01) and 0.63 at Bonneville Dam ($p$ = 0.001; Table 2 and Table B in S1 Appendix).

During 2008, survival estimates again indicated significantly higher survival for PIT-tagged fish at most detection locations, with AT/PIT ratios of 0.95 at Lower Monumental Dam ($p$ = 0.01), 0.91 at McNary Dam ($p$ = 0.095), 0.72 at John Day Dam ($p$ = 0.001), and 0.69 at Bonneville ($p$ = 0.021; Table 2 and Tables C and D in S1 Appendix). During both study years the only significant differences in travel time between tag treatment groups were at John Day Dam, where acoustic-tagged fish arrived 0.5 and 0.8 days later than PIT-tagged fish in 2007 ($p$ = 0.041) and 2008 ($p$ = 0.019), respectively (Figs A and B in S2 Appendix).

From yearling Chinook tagged and released during 2007, two fish with acoustic tags (0.001%) and 132 fish with PIT tags (0.003%) returned as adults to Bonneville Dam. Thus the

**Table 1. Characteristics of replicate release groups by life history type, migration year, and tag treatment.**

| Run type and migration year | Total fish released (n) | Mean size (range) | | Tag burden (%) | |
|---|---|---|---|---|---|
| | | Length (mm) | Weight (g) | Mean | Range |
| | | | Acoustic and PIT tag | | |
| Yearling Chinook | | | | | |
| 2007 | 3,818 | 133 (95-168) | 22.4 (8.3-46.6) | 3.1 | 1.5-8.9 |
| 2008 | 4,139 | 134 (92-202) | 23.1 (7.2-50.3) | 2.3 | 1.0-7.2 |
| Subyearling Chinook | | | | | |
| 2007 | 7,736 | 107 (95-146) | 12.8 (6.4–42.9) | 5.5 | 1.6-10.9 |
| | | | PIT tag only | | |
| Yearling Chinook | | | | | |
| 2007 | 46,714 | 133 (71-284) | — | — | — |
| 2008 | 50,814 | 136 (84-303) | — | — | — |
| Subyearling Chinook | | | | | |
| 2007 | 26,338 | 108 (82-158) | 13.8 (5.3-54) | 0.7 | 0.2-1.9 |

Tag burden defined as tag/fish weight in air; hyphens indicate groups tagged without weighing.

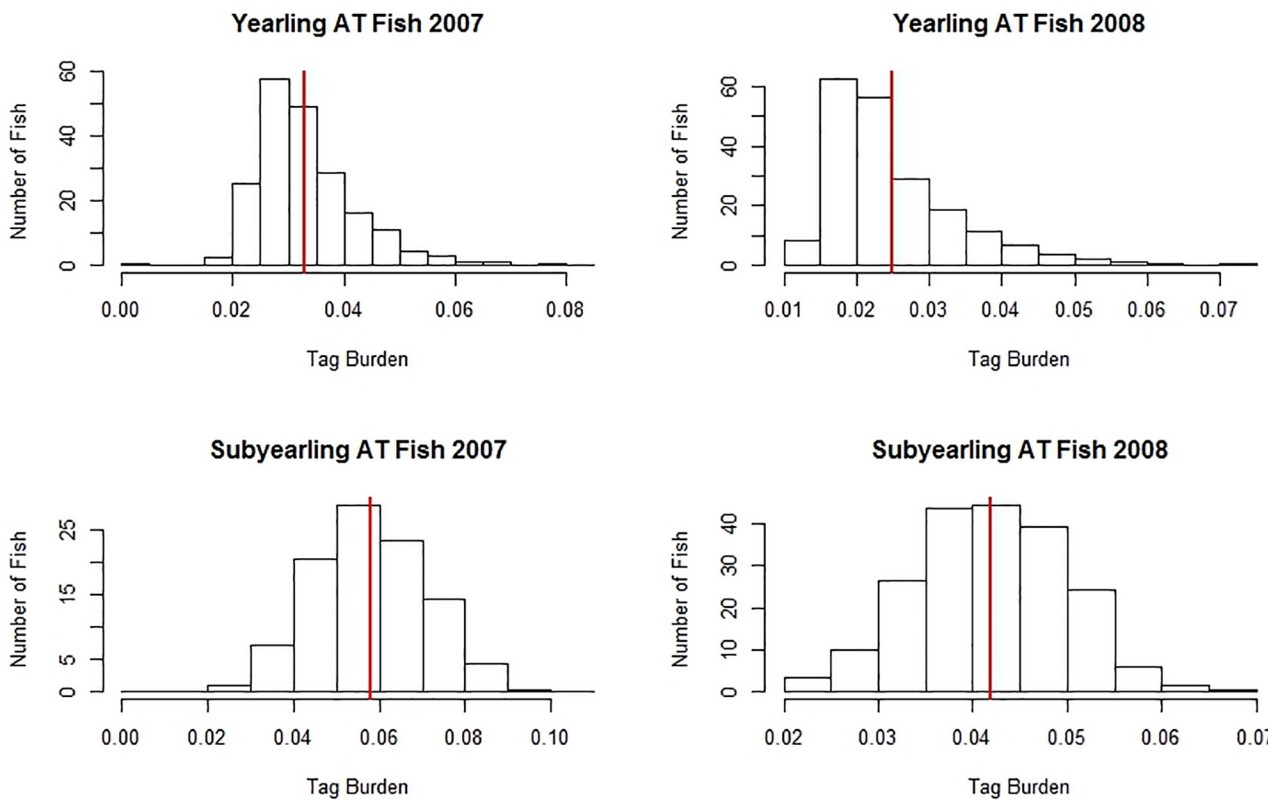

**Fig 2. Tag burden frequency distributions for acoustic-tagged (AT) study fish.** Tag burden is defined as the combined mass of the acoustic and PIT tags in air divided by the mass of the study fish in air.

ratio of smolt-to-adult returns for acoustic vs. PIT groups was 0.188 (SE 0.134), indicating significantly lower survival for acoustic-tagged fish ($p$ = 0.000). Adult returns of yearling fish were slightly higher for releases in 2008 than for those in 2007: a total of 23 fish with acoustic tags (0.006%) and 574 fish with PIT tags (0.011%) returned as adults to Bonneville Dam, resulting in a smolt-to-adult ratio of 0.424 (SE 0.046). While higher than in 2007, the smolt-to-adult ratio in 2008 remained significantly less than 1.0 ($p$ = 0.001).

**Subyearling Chinook salmon.** A total of 7,736 subyearling Chinook were surgically implanted with acoustic transmitters and PIT tags and released to the river in 2007 (Table 1). Additionally, 26,338 subyearlings were injected with PIT tags and released. Acoustic tagged fish, which also bore a PIT tag, experienced a combined mean tag burden of 5.5% (range 1.6-10.9%; Fig 2). For fish groups with only a PIT tag, mean tag burden was approximately 0.7% (range 0.2-1.9%). Subyearlings were grouped for analysis by week of release because insufficient numbers were detected after release to make daily comparisons of survival. Reliable estimates of survival for subyearlings were possible only at Little Goose and McNary Dam due to insufficient detections from acoustic-tag groups at all remaining locations.

Relative survival in terms of acoustic/PIT ratio was 0.80 at Little Goose Dam ($p$ = 0.003) and 0.41 at McNary Dam ($p$ < 0.001; Table 2). Mean travel time from release to Little Goose during 2007 was 1.2 d longer ($p$ = 0.000) for acoustic than for PIT-tagged fish (Fig C in S2 Appendix). The difference in travel time to McNary Dam was significant, with acoustic-tag groups arriving 2.7 d later that PIT-tag groups ($p$ = 0.002).

From subyearlings tagged and released in 2007, seven acoustic-tagged (0.001%) and 94 PIT-tagged fish (0.004%) returned to Bonneville Dam as adults. This produced a smolt-to-

**Table 2. Relative survival ratios of acoustic-tagged (AT) vs. passive integrated transponder (PIT) tagged Chinook salmon at each downstream detection site.**

| Detection site (dam) | Estimated survival | | Relative survival (AT/PIT) | t | P |
|---|---|---|---|---|---|
| | Acoustic tagged | PIT tagged | | | |
| | Yearling Chinook salmon 2007 | | | | |
| Little Goose | 0.93 (0.01) | 0.93 (0.01) | 1.0 | 0.14 | 0.893 |
| Lower Monumental | 0.92 (0.03) | 0.88 (0.01) | 1.05 | 1.98 | 0.080 |
| Ice Harbor | 0.81 (0.03) | 0.84 (0.01) | 0.99 | 1.14 | 0.285 |
| McNary | 0.72 (0.02) | 0.78 (0.01) | 0.92 | 2.21 | 0.054 |
| John Day | 0.62 (0.02) | 0.72 (0.02) | 0.86 | 3.25 | 0.010 |
| Bonneville | 0.50 (0.01) | 0.63 (0.04) | 0.79 | 4.87 | 0.001 |
| | Yearling Chinook salmon 2008 | | | | |
| Little Goose | 0.92 (0.01) | 0.95 (0.01) | 0.97 | 1.79 | 0.107 |
| Lower Monumental | 0.88 (0.02) | 0.93 (0.02) | 0.95 | 1.86 | 0.010 |
| Ice Harbor | 0.80 (0.02) | 0.83 (0.02) | 0.96 | 1.02 | 0.336 |
| McNary | 0.68 (0.03) | 0.75 (0.02) | 0.91 | 1.87 | 0.095 |
| John Day | 0.60 (0.04) | 0.83 (0.03) | 0.72 | 4.53 | 0.001 |
| Bonneville | 0.52 (0.03) | 0.75 (0.11) | 0.69 | 2.79 | 0.021 |
| | Subyearling Chinook salmon 2007 | | | | |
| Little Goose | 0.65 (0.05) | 0.81 (0.03) | 0.80 | 3.3 | 0.003 |
| McNary | 0.23 (0.02) | 0.56 (0.04) | 0.41 | 21.05 | <0.001 |

Standard errors are in parentheses; *t*-tests were derived from geomeans of AT/PIT survival ratios of paired replicates at each dam.

adult return ratio for AT/PIT fish of 0.250 (SE 0.098). Due to small sample size, the difference represented by this ratio was not statistically significant (*p* = 0.104).

## Gross necropsy, histology, and disease prevalence

**Yearling Chinook salmon.** We necropsied a combined total of 323 acoustic-tagged, 380 PIT-tagged, and 200 non-tagged reference fish for gross necropsy and microscopic examination of yearlings during 2007 and 2008. In general, reference fish appeared healthy on gross examination, with few abnormalities noted (Table B in S3 Appendix). In both acoustic- and PIT-tag treatment groups there appeared to be a general trend towards attrition of caecal and mesenteric fat reserves as fish migrated through the river, as well as increasing evidence of systemic inflammation. In comparisons between treatment groups migrating in 2008, there was significantly more food observed in the stomachs of PIT-tagged (18%) than acoustic-tagged (7%) fish arriving at Bonneville Dam (*p* = 0.038). A similar trend was observed during 2007, with food present in stomachs for 55% of PIT-tagged and 44% of acoustic-tagged fish; however, this difference was not significant.

All reference fish collected during both years appeared to be generally healthy on histological exam; therefore, we assumed no systematic bias to inter-treatment comparisons. Of the 42 metrics evaluated microscopically in 2007, significant differences between tag treatments were found in 7 from McNary Dam and 8 from Bonneville Dam (α = 0.01; Table B in S4 Appendix). In 2008, differences between treatments were seen in 5 of 49 metrics evaluated in fish recaptured at McNary Dam and in 8 of 48 metrics evaluated in recaptures at John Day/Bonneville Dam. Observed differences fell into four general categories: nutritional condition, peritoneal inflammation, infectious agents, and incision or injection-site healing (S4 Appendix).

Nutritional indices were mixed in direction, exhibiting no clear trend. However, for the most part, when significant differences were observed in metrics indicating peritoneal

inflammation, the scores were consistently higher (indicating more or greater inflammation) in acoustic-tagged than PIT-tagged fish. Although copious bacteria were not observed in the tissue sections examined, some tissue reactivity may have been elicited by secondary infection, introduced either during surgery or post-operatively through the incision.

Results from comparisons within the category of incision healing suggest that injection sites were cleaner and healed more quickly than surgical incisions. With respect to parasite load, a greater number of PIT fish had digenetic trematodes in the lower intestine at Bonneville/John Day Dam in 2008. However, all situations in which gastrointestinal trematode parasites were present were relatively minor, and there were no cases of significant host response (i.e. pathogenicity) associated with these parasites.

For yearling Chinook, ELISA values were mostly low in 2007 ranging from 0.068 to 0.131 in reference fish as well as in fish from both tag treatment groups (outliers 0.463 and 1.613). Therefore, no statistical analyses were conducted to compare antigen levels between treatments. In 2008, ELISA values were slightly higher, ranging 0.065-0.540 in reference fish and in fish from both tag treatments (outlier 1.11). At Bonneville Dam, 13% of acoustic- and 11% of PIT-tagged fish had moderate ELISA values. However, the difference between treatments was not significant at the $p \leq 0.050$ level.

**Subyearling Chinook salmon.** Survival rates for acoustic-tagged subyearlings in 2007 were extremely poor overall, and we were only able to recapture a total of nine fish from this treatment; thus, statistical power to identify differences among metrics was quite low. Downstream survival was significantly higher for PIT- than acoustic-tagged fish, and we were able to recapture 71 subyearlings from the PIT treatment. We also collected 79 non-tagged subyearlings at Lower Granite Dam in 2007 to serve as reference fish for evaluations of baseline health. As with yearling reference fish, subyearling reference fish appeared healthy overall on gross necropsy exam, and few abnormalities were noted (Table C in S3 Appendix).

Similar to findings from the yearling study, evidence of systemic inflammation appeared to increase as subyearling fish progressed downstream. Also similar was that caecal and mesenteric fat were rated higher in subyearling reference fish than in fish recaptured downstream. In comparisons between treatments, both caecal and mesenteric fat were also rated higher for PIT than acoustic-tagged fish; however, no statistically significant differences were found among the tag treatment groups in any metric evaluated by gross necropsy. This was primarily due to a very low sample size for acoustic-tagged fish.

We evaluated a total of 43 histological parameters or conditions for subyearlings recaptured in 2007 (Table C in S4 Appendix). Comparisons between tag treatments showed significant differences in 5 of the 43 metrics evaluated (α = 0.10). Similar to the comparisons of yearling fish, the majority of these differences fell into general categories describing peritoneal inflammation and healing at the incision or injection sites, with surgical incisions exhibiting more inflammation and injection sites healing faster (S4 Appendix).

Baseline levels of *Rs* antigen as measured by ELISA for subyearling reference fish ranged 0.070-0.213. At Bonneville Dam, combined ELISA values in recaptures from both tag groups ranged 0.078-0.442 overall and exceeded 0.299 in only two fish. Because these values were rated low for all but a few fish, no comparative analysis was conducted between tag treatment groups.

## Results from laboratory holding studies

### Yearling Chinook salmon

**Mean rates of survival.** For laboratory observations, we used 400 yearling Chinook salmon surgically implanted with both an acoustic transmitter and PIT tag in both study years.

Also in each year, 400 fish were injected with only a PIT tag, and 400 were only anesthetized and handled. In 2008, 400 additional fish were surgically implanted with PIT tags. Tag burdens were similar to those experienced by migrating fish of the same treatment type and year.

During 2007, yearling Chinook salmon exhibited a decline in survival over time (all treatments) throughout the 90-d holding period (Fig 3). For both tag treatment groups and for the reference groups, the downward slope of the survival function became more gradual after fish were transferred into seawater on day 15. For all treatment groups, mortality began to accelerate again after day ~56 of holding and then steadily increased through day 90.

At day 14 in 2007, mean survival for acoustic-tagged fish (0.85) was significantly lower than for PIT-tagged (0.92) and reference fish (0.93; $p = 0.027$). Likewise, at day 28, respective mean rates of survival for yearlings were 0.81, 0.89 and 0.89 in acoustic, PIT, and reference groups ($p = 0.012$; Table C in S5 Appendix). By day 90, mean survival among yearling treatment groups was not significantly different, at 0.64 for acoustic, 0.73 for PIT, and 0.74 for reference fish ($p = 0.159$). In 2008, a similar trend was observed, whereby the largest decline in survival for all study fish occurred during the freshwater holding period from day 0 through day 15 (Fig 3). No significant differences in survival were observed among tag treatments after day 14, 28, or 120 (Table D in S5 Appendix). During the freshwater holding phase in both years, survival was higher for yearlings held in the laboratory than for their respective PIT and AT cohorts migrating in the river (Fig 4). However, we did not evaluate whether these survival differences were statistically significant.

**Growth, tag loss, and disease prevalence.** Mean growth was measured by FL (mm) for pooled treatment groups surviving to the end of the holding period. Growth was not

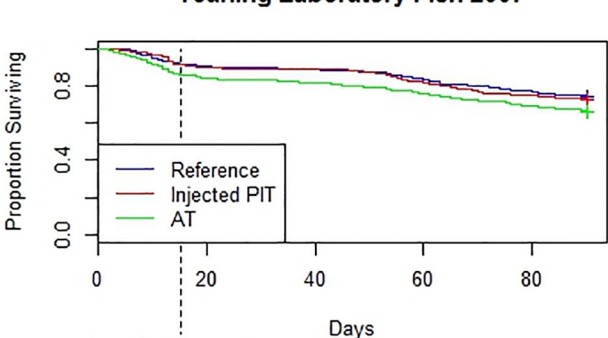
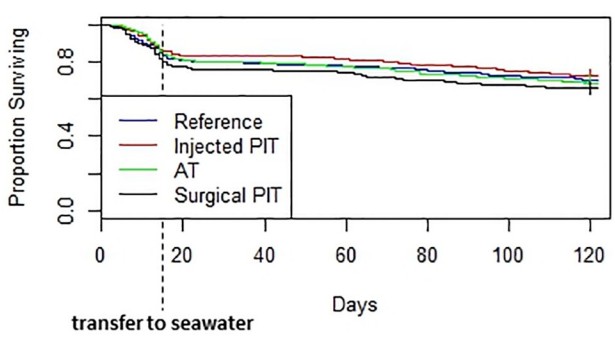
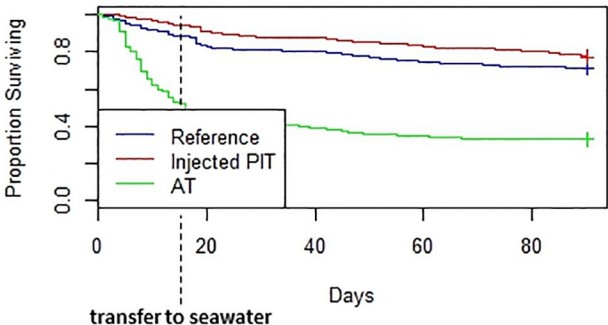
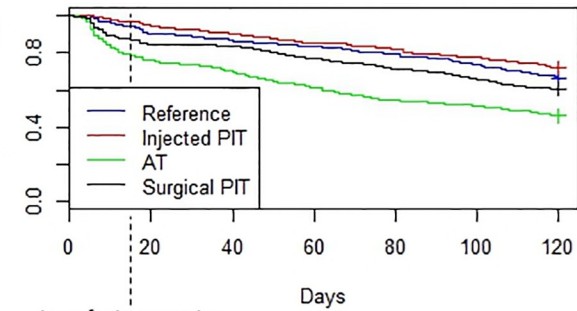

**Fig 3. Survival curves through day 90 in 2007 and day 120 in 2008 for study fish held in the laboratory.**

significantly different between tag treatments (Tables E and F in S5 Appendix). There was also no difference observed in weight gain (g) between treatments during 2007. Weights were not obtained from fish collected for PIT-tagging in 2007.

Of the yearlings that survived the 90-d holding period in 2007, five acoustic-tagged fish expelled or dropped PIT tags and one PIT-tagged fish expelled or dropped a tag. The difference in PIT-tag loss between treatments was small (1.7%; $p = 0.064$). Among acoustic-tagged survivors in 2007, none expelled or dropped an acoustic transmitter. Of fish surviving to the end of the 120-d holding period in 2008, one acoustic-tagged and one surgically implanted PIT-tagged fish expelled or dropped a PIT tag; however, 21 (8%) of acoustic-tagged fish expelled or dropped an acoustic transmitter. Acoustic tags were lost from fish of all replicates except the 8th and 10th. Timing of tag loss for these replicates is unknown because tags could not be observed and recovered from tanks without additional handling of fish prior to termination of the holding period.

Of fish held in 2007 and 2008 respectively, 334 and 480 died before termination of the study. For these fish, overall ELISA-coded values ranged 0.060-3.738. Samples taken from individual mortalities were averaged by replicate and treatment, and among ELISA-coded values from these samples, no significant differences were found in bacterial kidney disease levels among tag treatments in either year (Kruskal-Wallis $p = 0.774$ and 0.313). For the 814 and 1,095 fish that survived to termination of the study in 2007 and 2008, respectively, the range of ELISA values was similar (0.054-3.697). Mean coded values for individual ELISA samples were calculated by replicate and treatment, and no significant differences were found among treatments (Kruskal-Wallis $p = 0.993$ and 0.323).

## Subyearling Chinook salmon

**Mean rates of survival.** During 2007 and 2008 respectively, we surgically implanted 360 and 402 subyearlings with both acoustic and PIT tags, and we injected 360 and 408 with PIT tags only. We handled and anesthetized 360 subyearlings in 2007 and 405 in 2008 to serve as reference fish. In 2008, an additional 400 subyearlings were surgically implanted with PIT tags. Tag burdens experienced by fish in 2007 were similar to those of migrating cohorts with the same tag treatment (Table 1 and Fig 2). In 2008, fish implanted with both an acoustic and PIT tag experienced a mean tag burden of 4% (range 2-7%; Fig 2). Subyearlings injected or implanted only with a PIT-tag experienced a mean tag burden of 1% (range 0-1%).

During both study years, we observed a sharp decline in survival for acoustic-tagged subyearlings between days 0 and 18, after which comparative survival among treatment groups stabilized and remained fairly constant (Fig 3). In 2007, mean survival at day 14 was significantly different among treatments, at 0.53 for acoustic, 0.94 for PIT, and 0.88 for reference fish ($p = 0.000$; Table G in S5 Appendix), and was significantly lower for acoustic-tagged than for PIT-tagged and reference fish ($p < 0.001$). These differences were also significant at days 28 and 90 ($p = 0.000$).

In 2008, mean survival at day 14 was significantly different among all treatments, with estimates at 0.85 for acoustic, 0.97 for injected PIT, 0.88 for surgical PIT, and 0.94 for reference groups ($p = 0.037$; Table H in S5 Appendix). Survival was significantly lower for acoustic and surgically PIT-tagged groups than for injected PIT-tag and reference groups ($p = 0.044$). However, differences in survival among treatment groups were no longer significant at day 28 ($p = 0.827$) or day 120 ($p = 0.515$).

In general, survival for all treatments declined as tagging and holding temperatures increased from the first replicate collected to the last. Surgically tagged fish in particular appeared sensitive to warming temperatures. For example, day 14 survival among early vs. late

reference groups ranged 25% in 2007 and 24% in 2008. In comparison, day 14 survival among early vs. late acoustic-tag replicates ranged 79% in 2007 and 74% in 2008. In 2008, day 14 survival ranged over 78% among replicates of fish surgically implanted with PIT tags. However, the influence of water temperature on laboratory survival was not formally evaluated.

Similar to the pattern seen in yearling study fish, survival during the freshwater holding period was higher for both PIT and AT subyearlings than for their respective cohorts migrating in the river (Fig 4). Also similar to the yearling study fish, we did not evaluate whether these survival differences were statistically significant.

**Growth, tag loss, and disease prevalence.**   Mean growth in terms of FL (mm) and weight gain (g) was not significantly different for pooled treatment groups surviving the holding period in either 2007 or in 2008 (Tables I and J in S5 Appendix). For laboratory fish that survived the 90-d holding period in 2007, 3.4% (n = 4) of acoustic and no fish from PIT groups lost PIT tags ($p$ = 0.002). Acoustic transmitters were lost from a total of 9 (7.6%) acoustic treatment- fish, with losses occurring in replicates 11 (n = 4), 12 (n = 1), 16 (n = 1), 17 (n = 1) and 18 (n = 2). During 2008, no fish that survived to the end of the 120-d holding period expelled or dropped a PIT tag, whereas, 2% (n = 5) of acoustic-tagged fish dropped or expelled acoustic transmitters. Transmitters were lost from replicates 13 (n = 2) and 14, 16, and 20 (n = 1 each).

In 2007 and 2008, subyearlings that died prior to termination of laboratory holding totaled 695 and 677, respectively. In both years, overall ELISA values ranged 0.055-3.866 for these fish. During 2007, no significant difference in mean coded ELISA values was found among tag treatment groups (Kruskal-Wallis $p$ = 0.584). However, during 2008 ELISA values for subyearlings from reference and injected PIT groups were higher than those from surgical acoustic and surgical PIT groups (i.e. surgical PIT and surgical AT; Kruskal-Wallis $p < 0.001$).

For the 663 subyearlings that survived the holding period in 2007 and the 1,143 that survived in 2008, ELISA values ranged 0.040-3.441. No statistical analysis was conducted to evaluate differences among treatment groups in 2007 because with the exception of two fish, all values were low (0.040-0.240). No significant difference among treatment groups was found for survivors in 2008 (Kruskal-Wallis $p < 0.401$).

## Discussion

Results from this comprehensive, multi-faceted study provide strong evidence that for migrating juvenile salmon, surgical placement of both an acoustic and PIT-tag negatively affected survival and travel rates compared to injection with only a PIT-tag. Tag effects appeared to manifest differently over time and space, depending on the life stage and size of fish, as well as environmental conditions experienced during and after tagging. Although our laboratory studies provided insight into tag retention and incision healing, they also underestimated the effect of acoustic tagging, based on results from simultaneous field studies of migrating fish (Fig 4). These results highlight the importance of conducting tag effect studies in situ and under realistic conditions [16,18].

While we were not able to identify a single direct cause for the effects observed, we did identify key underlying factors that differentiated tag treatment groups through gross necropsy and histological examination. For example, the pathologies we identified in acoustic-tagged fish were consistent between years and among life histories and as such provided insight into how we can better craft protocols to minimize future effects of acoustic tagging. Acoustic-tagged fish were consistently more likely to exhibit an inflammatory response, both within the peritoneal cavity and at the incision site. This response may have placed a higher metabolic demand on acoustic-tagged fish and as such compromised their performance. An infectious cause for the inflammation observed in acoustic-tagged fish would have directly compromised

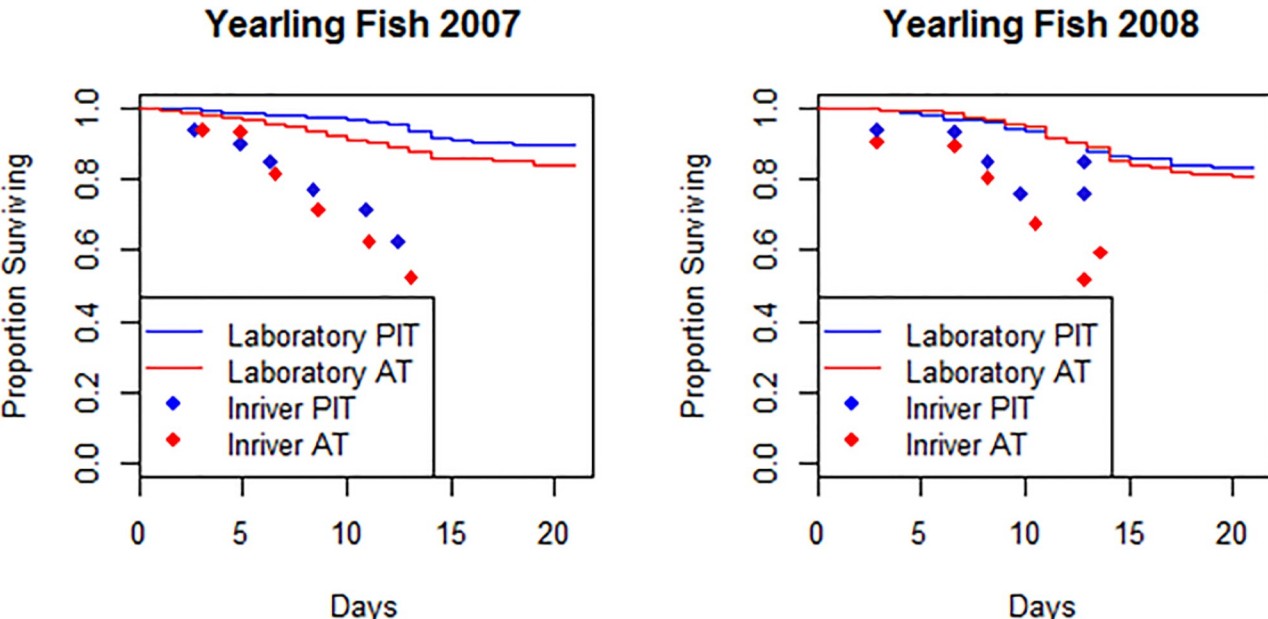

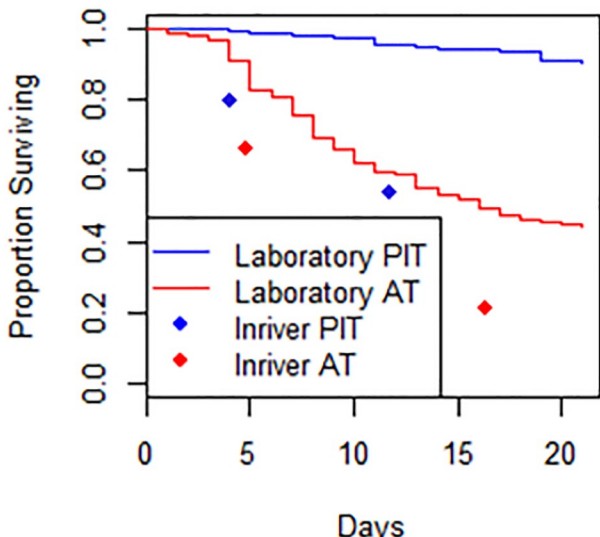

**Fig 4. Survival for AT and PIT fish held in the laboratory compared to those migrating in the river through day 20.**

the performance of these groups relative to PIT-tag treatment groups. However, it is also possible that inflammation was related to the thin layer of paraffin used to coat the acoustic tags to make them water-tight. In comparison with glass-encased passive integrated transponders, this paraffin coating may have contributed to a higher incidence of peritoneal inflammation, as suggested by Chisholm and Hubert [43] and Martin et al. [44].

Acoustic-tagged fish also exhibited delayed healing of the surgical incision site relative to fish with a PIT tag injection site. In histological comparisons, opposing tissue at injection sites

had better apposition than opposing tissue at surgical incision sites, allowing more rapid healing and restoration of a natural barrier to environmental pathogens. However, as mentioned above, no bacteria or fungi were observed within the peritoneal cavity during microscopic examination.

Finally, although not statistically significant, gross evaluation of caecal and mesenteric fat suggested that fish from acoustic tag groups were utilizing lipid reserves at a higher rate than those from PIT-tag groups. This may have occurred due to acoustic transmitter implants acting as mechanical appetite suppressants. Additionally, acoustic-tagged fish may also have been foraging less due to fatigue or morbidity, or they may have simply expended more energy during migration than PIT-tagged fish due to the added bulk and weight of their transmitters. Not surprisingly, physical tag burden, including both transmitter weight and bulk, has most often been described as being responsible for acoustic tag effects [45–51]. Carrying the transmitter can affect swimming ability or burst speeds, which can compromise the ability of fish to migrate, forage efficiently, and avoid predators. Relationships between body burden and energetics/performance in fish have been explored by others [46,52–55].

Overall, the observed differences in survival between treatments were more pronounced in small fish, further supporting a connection between tag effects and tag burden. Our covariate analysis of biological and physical factors experienced by migrating study fish indicated that aside from tag treatment, size of fish at tagging had the greatest influence on survival [6]. Relative survival of AT/PIT yearlings in 2007 approached 1.0 at fork lengths equal to or greater than 147 mm indicating no measurable effect on survival at tag burdens of less than approximately 2%. However, although mean tag burdens were as low as 1% for yearlings in 2008 and 1.6% for subyearlings in 2007, the difference in relative survival was observed in fish of all lengths.

Our study of subyearling fish also indicated that acoustic tag effects were amplified at higher temperatures. For example, during 2008, 14-d laboratory survival began to decline for treatment fish as water temperature at tagging approached and eventually exceeded 15˚C, however, observed declines were the most extreme in surgically tagged groups. For these fish, survival to day 14 was lowest for the last replicate tagged (replicate 10), for which temperature was 17˚C during tagging and approached 20˚C during freshwater holding. These results highlight the potential for substantial tag effects when surgical implantation is conducted at temperatures in excess of 15˚C, and we recommend caution at or above these temperatures.

This recommendation is similar to guidelines already in place for less invasive tagging methods. For example, a maximum temperature threshold of 17˚C is recommended for PIT-tagging in field manuals of both the Columbia Basin Fish and Wildlife Authority [56], and the Bonneville Power Administration [57]. Both field manuals include a cautionary statement that this threshold should be lowered under circumstances where additional stressors may be present, and the Bonneville manual stated that "as temperature increases above 15˚C, fish become stressed very easily" [56]. Based on our observations, acoustic-tagged fish did experience additional stressors compared to PIT-tagged fish, from the presence of the transmitter as well as from the surgical procedure.

When surgical tagging must take place during relatively warm water conditions, another factor to consider is whether or not to close the incision. In 2008, we conducted experiments with surgically PIT-tagged subyearlings to evaluate whether a subset of chemicals used in aquaculture would be efficacious in preventing or minimizing inflammatory reactions and possible infection at incision sites [6]. Results from these analyses indicated that holding temperatures, the presence of 2 secure ligatures (compared to 0 or 1), and the presence of foreign material on sutures were predictive of survival to day 28 for fish surgically implanted with PIT tags. Higher

post-surgical mortality for fish with sutures present at day 7 or later may have occurred due to secondary infection from pathogens that accumulated on sutures such as fungi or bacteria.

When foreign matter had accumulated on sutures, there was often evidence at gross exam of secondary dermal ulceration directly beneath the mass of foreign material. These ulcers could have facilitated and served as a route for internal infection. During their study to evaluate the effects of PIT tags in juvenile Atlantic salmon (*Salmo salar*), Larson et al. [9] also observed that incisions closed with sutures were prone to fungal infection, while those without sutures healed well. They reported that such infection was a particular problem in smaller fish, and ultimately concluded that sutures should not be used in fish measuring less than 135 mm FL. The idea that suture presence contributes to tag effects also arose during development studies of the micro-acoustic tags used in our studies [58,59]. Our findings supported such a correlation and have since prompted the development of an injectable acoustic transmitter [60].

Nevertheless, injectable transmitters will not be appropriate for every situation. Where surgery is required, rapidly dissolving sutures may be used (i.e. braided or finer/lighter suture than typically used) or sutures may be foregone altogether to reduce anticipated inflammation and infection at the incision site [9,10,18]. Higher risk of infection by braided sutures "wicking" contaminates into the body cavity will need to be balanced against the greater likelihood of tag loss from the incision site if using fine or no suture material. Another alternative is to hold surgically tagged fish until their incisions heal, although, removal of residual sutures would require a second round of handling. Also, fish held for long periods after tagging may not experience the same migration conditions as cohorts from the populations they are intended to represent.

With respect to transmitter retention, our laboratory results suggested that acoustic tag loss may inflict bias in analyses of data from field studies when a secondary mark, such as a PIT tag, is not available. A secondary mark was important in helping us to distinguish acoustic transmitter loss from mortality. We were unable to determine timing of tag loss for the majority of study fish; however, reports of transmitter loss in fish are well documented. Tag loss is most often reported as an active expulsion that occurs several days to weeks after surgery rather than a passive loss through an unhealed incision [10,18,43,45,50,61–63].

Holding studies also indicated that mortality related to surgical acoustic tagging had run its course by day 15 for yearlings and by day 18 for subyearlings. However, smolt-to-adult return ratios for migrating study fish indicated that a tag-based survival differential continued to increase well beyond the freshwater migration period. The freshwater migration was 13 d on average for yearlings and would have been significantly longer for subyearlings, had enough survived to provide a meaningful estimate. We do know that migrating subyearlings took 14.5 d on average to travel the 225 km from Lower Granite to McNary Dam, about half the total distance required to pass Bonneville.

Finally, overall, ELISA values indicate that prevalence of bacterial kidney disease was low, and we found no significant differences among treatment groups in most comparisons. Although bacterial kidney disease tends to be chronic in nature, there was some indication that it contributed to mortality in PIT-tag only subyearlings during 2008. As a group, subyearling Chinook held in the lab during 2008 had a higher prevalence of the *Rs* antigen than yearlings held in the same year or than subyearlings held in the previous year. Furthermore, injected subyearlings that died in 2008 were found to have even higher ELISA values than surgically implanted subyearlings with either PIT or acoustic tags. We also identified a trend in 2008 toward higher mean ELISA values for fish that died progressively later in the holding period. One possible explanation for these results is that mortality for injected treatments was driven by bacterial kidney disease, while mortality in surgical treatments, which tended to occur within the first few weeks of holding, was more related to the effects of surgery.

## Conclusion

The work that has gone into downsizing and developing biologically inert telemetry transmitters is admirable, and the fisheries research community in general has come a long way with respect to developing safe tagging techniques and protocols. On both fronts, we have absolutely moved closer towards the ultimate goal of "no measureable impact" to tagged study fish. However, we need to remain cognizant that we are not yet there for all fish under all conditions and that the gold standard of "no effect" may not be achievable.

Simply handling fish in the most benign way can alter behavior, condition, and performance and can influence rates of survival. Results from our study showed juvenile Chinook salmon in general experiencing a decline in condition as they moved seaward. Additional stressors placed on acoustic-tagged fish, such as increased body burden, introduction of a foreign body, and the presence of a surgical skin incision, were enough to influence their survival and behavior in a measureable way.

Continued downsizing of transmitters, use of biocompatible materials, and tagging methods that are minimally invasive (e.g. injection) are all worthwhile endeavors. However, such advances in miniaturization also provide the impetus for researchers to literally "push the envelope," by injecting tags into fish that are simply too small to accommodate these implants. Therefore, we recommend that researchers in general adopt a more *skeptical* viewpoint when designing, conducting, or reviewing tagging studies, regardless of the tag size or the tagging methods available.

Results of our study provide insight into the types of tag effects one should anticipate when conducting and/interpreting the results of telemetry studies that rely on internal devices. General references to guide researchers in surgical or other implantation techniques are readily available [64–66]. Unfortunately, there appears to be no blanket set of rules that can be applied universally to ensure representative survival or behavior: in every tagging study, assumptions regarding the validity of extrapolations must be carefully considered.

## Supporting information

**S1 Table. Tag specifications.** Specifications of acoustic tags (Juvenile Salmonid Acoustic Telemetry System transmitters, or JSAT) and passive integrated transponder tags (SST tag TX-1411SST) used in both study years.
(DOCX)

**S1 Appendix. Downstream survival.**
(DOCX)

**S2 Appendix. Travel time.**
(DOCX)

**S3 Appendix. Gross necropsy.**
(DOCX)

**S4 Appendix. Histological evaluation.**
(DOCX)

**S5 Appendix. Laboratory holding study.**
(DOCX)

**S6 Appendix. Evaluation of disease prevalence.**
(DOCX)

## Acknowledgments

This work would not have been possible without the following researchers: Lyle G. Gilbreath, Mark S. Myers, Mark E. Peterson, Lila L. Charlton, Steven G. Smith, Tiffany Marsh, Lynn McComas, Niel Paasch, Ken McIntyre, Jim Simonson, John Ferguson, and Gene M. Matthews.

## Author Contributions

**Conceptualization:** A. Michelle Wargo Rub, Benjamin P. Sandford.

**Data curation:** A. Michelle Wargo Rub, Benjamin P. Sandford, April S. Cameron.

**Formal analysis:** A. Michelle Wargo Rub, Benjamin P. Sandford.

**Funding acquisition:** A. Michelle Wargo Rub.

**Investigation:** A. Michelle Wargo Rub.

**Methodology:** A. Michelle Wargo Rub, Benjamin P. Sandford.

**Project administration:** A. Michelle Wargo Rub.

**Resources:** A. Michelle Wargo Rub.

**Software:** Benjamin P. Sandford.

**Supervision:** A. Michelle Wargo Rub.

**Validation:** A. Michelle Wargo Rub, Benjamin P. Sandford, April S. Cameron.

**Writing – original draft:** A. Michelle Wargo Rub, Benjamin P. Sandford.

**Writing – review & editing:** A. Michelle Wargo Rub, Benjamin P. Sandford, JoAnne M. Butzerin, April S. Cameron.

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
