## [Decision Letter · Decision Letter 0]

2 Oct 2019

PONE-D-19-17402

Pushing the envelope: micro transmitter effects on small juvenile Chinook salmon (Oncorhynchus tshawytscha)

PLOS ONE

Dear Dr. Wargo Rub,

Thank you for submitting your manuscript to PLOS ONE. After careful consideration, we feel that it has merit but does not fully meet PLOS ONE’s publication criteria as it currently stands. Therefore, we invite you to submit a revised version of the manuscript that addresses the points raised during the review process.

Two reviewers have provided minor edits/changes which would enhance the manuscript. Overall this study is well executed and well written and it should not take much effort to make the changes outlined in the reviews.

We would appreciate receiving your revised manuscript by Nov 16 2019 11:59PM. To enhance the reproducibility of your results, we recommend that if applicable you deposit your laboratory protocols in protocols.io, where a protocol can be assigned its own identifier (DOI) such that it can be cited independently in the future. For instructions see: http://journals.plos.org/plosone/s/submission-guidelines#loc-laboratory-protocols

We look forward to receiving your revised manuscript.

Kind regards,

Madison Powell, PhD

Academic Editor

PLOS ONE

Journal Requirements:

2. In your Methods section, please provide additional location information of the study areas, including geographic coordinates for the data set if available.

'NOAA Fisheries, NWFSC, Fish Ecology Division (AMWR, BPS, and JMB) received

funding for this study from The Environmental Resources Branch, Planning and

Engineering Division

Portland District, U.S. Army Corps of Engineers, Robert Duncan Plaza

333 S.W. 1st Avenue, Portland, Oregon 97208-2946 under contract #

W66QKZ60441152

The funding agency had no role in the study design, data collection and analysis,

decision to publish, or preparation of the manuscript.'  

We note that one or more of the authors are employed by a commercial company: Ocean Associates, Inc.

Additional Editor Comments (if provided):

This manuscript is well written and focuses on a complex aspect of transmitter effects on fish. In this case Chinook salmon are studied and the results provide data of interest to fisheries and conservation biologists studying this species. The data are also applicable to other salmonids to a degree. Two reviewers have provided only minor corrections which would enhance the clarity and readability of the manuscript.

Reviewers' comments:

Reviewer's Responses to Questions

**Comments to the Author**

1. Is the manuscript technically sound, and do the data support the conclusions?

Reviewer #1: Yes

Reviewer #2: Yes

2. Has the statistical analysis been performed appropriately and rigorously? 

Reviewer #1: Yes

Reviewer #2: Yes

3. Have the authors made all data underlying the findings in their manuscript fully available?

Reviewer #1: Yes

Reviewer #2: Yes

4. Is the manuscript presented in an intelligible fashion and written in standard English?

Reviewer #1: Yes

Reviewer #2: Yes

5. Review Comments to the Author

Reviewer #1: 1. On Figure 3 (survival curve for lab experiment) place an arrow on the date when the fish were transferred to seawater. Doesn’t have to be exactly that but I think it should be evident on the graph when the seawater transfer was made.

2. The major finding of the work is that acoustic tagging exacts a greater mortality penalty for free-swimming fish than occurs for fish held in laboratory conditions. It would be great to have a figure that clearly illustrates this. Table 2 shows survival for free-living fish based on location (distance from release site), Figure 3 shows the mortality curve (in days) for laboratory fish. Could the data from the two be combined to directly show mortality curves for both groups of fish? For free-swimming fish, could use travel time (days) to a given location for the x-axis. This would be an additional figure.

3. The author’s don’t demonstrate a clear cause for mortality in the free-swimming fish, nor do they find much evidence of serious deficiency in the laboratory fish. That’s fine, their work is a solid accomplishment. Their discussion of mechanism is found in a number of very brief paragraphs lines 529 – 572. This is appropriate, as the author’s found little evidence for direct tagging affects that would cause mortality but it is also a bit unsatisfying. I would suggest a larger, more cohesive paragraph that summarizes their results and concludes that they’ve found little evidence for a direct mechanism causing mortality.

Reviewer #2: The authors provided a detailed and comprehensive evaluation of the effect of internal tagging on wild fish welfare as they migrate. This was followed up with laboratory comparisons. As with any study performed in the field it is difficult to sometimes get sufficient and exact numbers but the authors have accurately presented their findings and are quick to note where this findings were significant or not and to make note of general trends of the information which is important in this kind of study.

6. PLOS authors have the option to publish the peer review history of their article (what does this mean?). If published, this will include your full peer review and any attached files.

Reviewer #1: No

Reviewer #2: No

---

## [Author Response · Author response to Decision Letter 0]

29 Oct 2019

October 25, 2019

Dear Dr. Powell,

Thank you for considering our manuscript “Pushing the envelope: micro transmitter effects on small juvenile Chinook salmon (Oncorhynchus tshawytscha)” for publication in PLOS ONE. We have revised our original manuscript in response to the points raised by yourself and two outside reviewers. Our response to each reviewer’s comments/requests is included below. 

We hope you will find that our revised manuscript meets PLOS ONE’s standards for publication.

Sincerely,

A. Michelle Wargo Rub 

Editors Comments/Requests:

Response: We have made every attempt to ensure that our manuscript meets PLOS ONE’s style requirements.

2. In your Methods section, please provide additional location information of the study areas, including geographic coordinates for the data set if available.

Response: We have expanded our description of the study area and have included geographic coordinates for the locations where fish collection/tagging and recapture occurred. 

Revised text (p. 4; lines 72-76): Field studies were conducted at multiple locations spanning several hundred km within the Columbia River Basin in the U.S., Pacific Northwest (Fig 1). Fish were collected, tagged, and released at Lower Granite Dam on the Snake River (rkm 695; 46.6604°N, 117.4280°W) and monitored as they migrated downstream to Bonneville Dam (rkm 235; 45.6443°N, 121.9406°W).

We also included gps coordinates in the caption for Figure 1 for the dams where fish were recaptured along their migration. 

Revised text (p. 5; lines 84-89): Fig 1. Columbia River hydropower system, Pacific Northwest United States. Fish were collected, tagged, and released at Lower Granite Dam (46.6604°N, 117.4280°W). Diamonds indicate downstream detection sites for acoustic transmitters and circles for PIT tags. Separation by Code systems were used to recapture Chinook salmon smolts at McNary Dam (45.9362°N, 119.2972°W), John Day (45.7148°N, 120.6937°W), and Bonneville Dam (45.6443°N, 121.9406°W).

3. Revised Financial Disclosure statement.

NOAA Fisheries, NWFSC, Fish Ecology Division (AMWR, BPS, and JMB) received funding for this study from Portland District, U.S. Army Corps of Engineers, under contract W66QKZ60441152. This agency provided support in the form of salaries for AMR & BPS but did not have any role in the study design, data collection and analysis, decision to publish, or preparation of the manuscript.

Funding in the form of salary for ASC was provided by NOAA Fisheries through contract with Ocean Associates, Inc. Ocean Associates, Inc. had no role in the study design, data collection and analysis, decision to publish, or preparation of the manuscript. 

The specific role of each author is articulated in the ‘author contributions’ section. 

Neither the affiliation with the U.S. Army Corps of Engineers nor with Ocean Associates, Inc. altered the authors' adherence to PLOS ONE policies on sharing data and material.

Reviewer’s Comments/Requests:

Reviewer #1: 

Thank you for your thoughtful suggestions. We appreciate your time and input and have attempted to address all three of your requests. We believe the manuscript is improved as a result. 

1. On Figure 3 (survival curve for lab experiment) place an arrow on the date when the fish were transferred to seawater. Doesn’t have to be exactly that but I think it should be evident on the graph when the seawater transfer was made.

Response: Great suggestion- we have added a dashed line on each plot in Figure 3 to indicate seawater transfer.

2. The major finding of the work is that acoustic tagging exacts a greater mortality penalty for free-swimming fish than occurs for fish held in laboratory conditions. It would be great to have a figure that clearly illustrates this. Table 2 shows survival for free-living fish based on location (distance from release site), Figure 3 shows the mortality curve (in days) for laboratory fish. Could the data from the two be combined to directly show mortality curves for both groups of fish? For free-swimming fish, could use travel time (days) to a given location for the x-axis. This would be an additional figure.

Response: We think this is also a great suggestion and have added Figure 4 in response. Figure 4 combines laboratory survival curves through day 20 for AT and PIT fish (yearlings 2007 and 2008 and subyearlings 2007) and point estimates of survival for AT and PIT fish migrating in the river by median travel time to each downstream detection location. 

3. The author’s don’t demonstrate a clear cause for mortality in the free-swimming fish, nor do they find much evidence of serious deficiency in the laboratory fish. That’s fine, their work is a solid accomplishment. Their discussion of mechanism is found in a number of very brief paragraphs lines 529 – 572. This is appropriate, as the author’s found little evidence for direct tagging affects that would cause mortality but it is also a bit unsatisfying. I would suggest a larger, more cohesive paragraph that summarizes their results and concludes that they’ve found little evidence for a direct mechanism causing mortality.

Response: We now state in the discussion that we were not able to identify a single direct cause for the tag effects observed (p. 27; lines 533-535). We also reorganized the discussion to better highlight the key pathologies we identified in the acoustic-tagged fish that serve to provide insight into how tagging protocols might be crafted in the future to minimize/alleviate tag effects. We believe these changes have improved the discussion and hope these efforts have addressed the reviewer’s concerns. 

Revised text (p. 27; lines 532-534): While we were not able to identify a single direct cause for the effects observed, we did identify key underlying factors that differentiated tag treatment groups through gross necropsy and histological examination.

Reviewer #2: The authors provided a detailed and comprehensive evaluation of the effect of internal tagging on wild fish welfare as they migrate. This was followed up with laboratory comparisons. As with any study performed in the field it is difficult to sometimes get sufficient and exact numbers but the authors have accurately presented their findings and are quick to note where this findings were significant or not and to make note of general trends of the information which is important in this kind of study.

Thank you sincerely for your time and support.

---

## [Decision Letter · Decision Letter 1]

24 Feb 2020

Pushing the envelope: micro transmitter effects on small juvenile Chinook salmon (Oncorhynchus tshawytscha)

PONE-D-19-17402R1

Dear Dr. Wargo Rub,

We are pleased to inform you that your manuscript has been judged scientifically suitable for publication and will be formally accepted for publication once it complies with all outstanding technical requirements.

With kind regards,

Madison Powell, PhD

Academic Editor

PLOS ONE

Additional Editor Comments (optional):

The manuscript provides worthwhile information and all previous comments from reviewers have been addressed.

Reviewers' comments:

Reviewer's Responses to Questions

**Comments to the Author**

1. If the authors have adequately addressed your comments raised in a previous round of review and you feel that this manuscript is now acceptable for publication, you may indicate that here to bypass the “Comments to the Author” section, enter your conflict of interest statement in the “Confidential to Editor” section, and submit your "Accept" recommendation.

Reviewer #2: All comments have been addressed

2. Is the manuscript technically sound, and do the data support the conclusions?

Reviewer #2: (No Response)

3. Has the statistical analysis been performed appropriately and rigorously? 

Reviewer #2: (No Response)

4. Have the authors made all data underlying the findings in their manuscript fully available?

Reviewer #2: (No Response)

5. Is the manuscript presented in an intelligible fashion and written in standard English?

Reviewer #2: (No Response)

6. Review Comments to the Author

Reviewer #2: (No Response)

7. PLOS authors have the option to publish the peer review history of their article (what does this mean?). If published, this will include your full peer review and any attached files.

Reviewer #2: No

---

## [Editor Report · Acceptance letter]

11 Mar 2020

PONE-D-19-17402R1 

Pushing the envelope: micro‑transmitter effects on small juvenile Chinook salmon (*Oncorhynchus tshawytscha*) 

Dear Dr. Wargo Rub:

I am pleased to inform you that your manuscript has been deemed suitable for publication in PLOS ONE. Congratulations! Your manuscript is now with our production department. 

With kind regards,

on behalf of

Dr. Madison Powell 

Academic Editor

PLOS ONE